# Antioxidant Properties of Dried Ginger (*Zingiber officinale* Roscoe) var. *Bentong*

**DOI:** 10.3390/foods12010178

**Published:** 2023-01-01

**Authors:** Iswaibah Mustafa, Nyuk Ling Chin

**Affiliations:** 1School of Chemical Engineering, College of Engineering, Universiti Teknologi MARA Cawangan Terengganu, Bukit Besi Campus, Dungun 23200, Terengganu, Malaysia; 2Department of Process and Food Engineering, Faculty of Engineering, Universiti Putra Malaysia, Serdang 43400, Selangor, Malaysia

**Keywords:** drying, extraction solvent, *Bentong* ginger, antioxidants

## Abstract

Ginger (*Zingiber officinale* Roscoe) is a popular culinary herb used in the Eastern culture. The essential cultivar of the Zingiber genus is rich in antioxidants and is crucial in the fight against oxidative stress-related diseases. The antioxidant properties of dried ginger were evaluated and compared for their efficacy from different drying processes (sun-, oven-, vacuum- and freeze-drying) and using three extraction solvents: hot water, aqueous ethanol (80%, *v*/*v*) and ethanol. The drying process demonstrated a positive effect on the antioxidant activities of ginger. A significant difference (*p* < 0.05) was observed in the extracting ability of each solvent. Sun-dried ginger extracted with ethanol performed better than the fresh ginger extract in the form of increased yield (3.04-fold), TFC values (12.25-fold), reducing power (FRAP) (15.35-fold), total antioxidant activity (TAA) (6.82-fold) and inhibition of ABTS^•+^ radical cation (3.51-fold) and DPPH^•^ radical (95%). Meanwhile, freeze-dried aqueous ginger extracts demonstrated significantly higher TPC (1.66-fold), TFC (3.71-fold), FRAP (3.26-fold), TAA (2.97-fold), ABTS^•+^ scavenging activity (1.48-fold) and DPPH^•^ radical inhibition (77%), compared to fresh ginger extracts. In addition, it was found that ethanol was significantly superior to aqueous ethanol in phenolic content recovery, despite the lower yield. Furthermore, ethanol ginger extracts exhibited higher antioxidant activity than aqueous ethanol extracts. On the other hand, hot water was the least potent solvent for extraction. In summary, there was an excellent correlation between TPC, TFC and antioxidant activity. Sun-drying is the most desirable method for preserving and enhancing ginger quality due to its cost effectiveness and bioactive compound efficacy.

## 1. Introduction

Ginger is widely grown in Asia, Europe and the Middle East for its desirable properties as an herb or spice, and is also used in folk medicine. *Bentong* ginger (*Zingiber officinale* Roscoe) is known as the “King of Ginger” and is the most sought-after variety cultivated in highland areas with crystal clear waters and fertile soil of the Bentong region (3.5222° N, 101.9104° E) because of the superior taste and nutritional value compared to other varieties. It is horn-shaped, pale yellowish, branched, and has scaly structures with a spicy lemon-like scent [1]. *Bentong* ginger has received geographical indication by the Malaysian government, and it possesses specific characteristics based on its origin. The ginger production in Malaysia increased significantly from 2001 to 2019 to approximately 11,205 metric tons per year (Department of Agriculture of Malaysia). This improvement is a 100% increase in market demand from 2014 to 2018, indicating the rising health awareness among consumers [2]. 

Ginger has been the focus of interest, especially in the Eastern region, because the plant is rich in potent antioxidants known as polyphenol compounds. These compounds act as an active phytochemical to prevent significant oxidation-linked diseases, such as cancer promotion, arthritis, cognitive diseases, diabetes mellitus, and cardiovascular diseases [3]. Several studies have explored the bioactive compounds in ginger and their antioxidant activities [4,5,6]. According to Carocho and Ferreira [7], antioxidant compounds rebalance the production of free radicals, subsequently interrupting the oxidative stress mechanism [7]. Besides the extensive medicinal purposes, antioxidants also play a role in food preservation by extending the shelf life of food products [8]. In addition, Tohma et al. [8] stated that the most studied compounds in ginger are phenolic compounds, acid-phenols and the antioxidant activity of flavonoids [8]. Apart from that, ginger has been proven as a promising source of health-promoting compounds, such as essential oil, minerals, vitamins and fibers [9,10,11].

Fresh ginger is classified as a highly perishable commodity, like other fresh fruits and vegetables, that deteriorates quickly within a short time due to its high moisture content. Fresh ginger contains 78.89% moisture [12] and should be stored in a dry condition at low temperatures to maintain the maximum nutritional value. Other preservation methods should be adopted when these conditions are not met to prevent the spoilage of the ginger. However, it is well reviewed that processing conditions have notable effects on the bioactive constituents of plant materials, particularly anthocyanins, carotenoids, phenolic antioxidants, and ascorbic acid (AA) [13]. 

Drying is one of the most common preservation methods that remove water from fresh produce. This method can stop microbial growth and prevent biochemical modifications. Furthermore, moisture reduction could also imply a significant decrease in weight and volume, transportation and storage costs, minimizing packaging, and food consumption during the off-season [14]. Nevertheless, dried food experiences shrinking, an undesirable quality in fresh food [5]. Shrinking happens due to unbalanced pressure resulting from water removal, thus inducing stress and changing the food cellular structure to become more porous for higher water adsorption [15]. 

Sunlight is an abundant, inexhaustible and non-polluting essential renewable energy [16]. Therefore, sun-drying is a sustainable processing method that has become increasingly popular since the technique does not require high investment and maintenance in operation [16,17]. It has also been reported that sun-drying contains ultraviolet-B (UVB) rays, which positively impacts the antioxidant composition and activity in dried products [6,18,19]. This technique is convenient in many tropical and subtropical regions. For instance, Malaysia experiences a hot and humid climate throughout the year with an average daily temperature of 21–32 °C, which offers a favorable condition to produce and preserve ginger sustainably and cost effectively. 

Therefore, this study aimed to assess and compare the antioxidant properties of *Bentong* ginger when different drying methods are used, i.e., naturally using sun-drying and equipment-based drying practices, which include freeze-, oven- and vacuum-drying. In addition, the objective of this research is to examine the antioxidant properties and sustainable yield of *Bentong* ginger extracts and determine the best drying method that is environmentally friendly using green solvents such as hot water and ethanol.

## 2. Materials and Methods

### 2.1. Sample Preparation

*Bentong* ginger was selected for this study because it is the most demanded ginger variety among consumers in Malaysia. A total of 1 kg of fresh ginger rhizomes with uniform size, color, and maturity stage was harvested from a farm in Kampung Baru, Bentong, Malaysia. Then, the ginger samples were washed to remove soil and contaminants and manually sliced into less than 5 mm thickness before being subjected to different drying treatments. Consistent cutting was observed to ensure equal and consistent heat distribution when drying the ginger slices. Fresh ginger was used as a control sample. 

### 2.2. Drying of Ginger

Drying was carried out based on Mustafa et al. [6]. The ginger was subjected to drying in a convection oven (UM500, Memmert GmbH Schwabach, Germany) and vacuum oven (Jeiotech OV11, Lab Companion, UK) at 60 °C at less than 30 cm Hg pressure for three days following the work of Chumroenphat et al. [20] and Ding et al. [21]. According to Ding et al. [21], a mild temperature of 60 °C was recommended for the appearance and formation of ginger compounds. Additionally, Chumroenphat et al. [20] demonstrated that the highest antioxidant capacity for ginger was obtained at 60 °C, and any higher drying temperature would reduce antioxidant capacity [20]. 

Freeze-dried ginger was prepared by freezing the samples at −30 °C in a freezer (Haier, Biomedical, Puchong, Malaysia) and lyophilized for three days in a freeze dryer (Coolsafe Benchtop, Scanvac, Sweeden) until completely dry. Then, for sun-drying, ginger slices were laid and spread on a 70 cm diameter round rattan tray and dried under direct sunlight at temperatures between 28 and 44 °C for three days. The samples were exposed to approximately 36 h of daylight, with a mid-day temperature reaching up to 44 °C. Moisture content was determined using a moisture analyzer (MX-50, A&D Company, Tokyo, Japan). After the final moisture content of the ginger slices was reduced to below 10–20% with 80–90% dry matter content [14], the samples were finely ground into a powder form using an electrical food blender (RT-02A, Taichung, Taiwan). The final moisture content was determined based on the microbial stability in the product. Spoilage-causing microorganisms’ activities and survival are significantly reduced when the moisture content is below 10% [22]. Subsequently, the powdered samples were packed in an air-tight container and stored at 4 °C (Model SD-700, Protech, Balakong, Malaysia) until further analysis. 

### 2.3. Antioxidant Analysis 

#### 2.3.1. Extraction of Ginger Bioactive Compounds Using Different Solvents

Fresh and dried ginger were extracted using three different solvents: hot water, aqueous ethanol (80%) and ethanol (100%), following a ratio of 1:10 [23]. The maceration was conducted in conical flasks using a shaker at 150 rpm for 24 h at room temperature (Wisecube WIS-30, Daihan Scientific, Gangwon-do, Korea). The extracts were then filtered through a Fioroni Grade 601 filter paper to separate the liquid. The process was repeated using the residual biomass and the respective solvents to ensure complete extraction. After that, the extracts were allowed to evaporate at 40 °C to yield dry solid extracts. The dried extracts were later weighed to determine the yield percentage using Equation (1). Next, the dried extracts were prepared as standard stock solutions in the respective solvents at a 1 mg/mL concentration and stored at 4 °C until further use.
(1)$$\mathrm{Yield}\;\%=\frac{\mathrm{Weight}\;\mathrm{of}\;\mathrm{dry}\;\mathrm{extract}}{\mathrm{Weight}\;\mathrm{taken}\;\mathrm{for}\;\mathrm{extraction}\;}\;\times \;100$$

#### 2.3.2. Determination of Total Phenolic Content (TPC)

The TPC of ginger extracts was measured according to the Folin–Ciocalteu method [24]. First, distilled water was added to the ginger extract solution (100 µL) to make a volume of 1 mL. Then, 0.5 mL of the Folin–Ciocalteu reagent (diluted 1:1 with water) and 2.5 mL (20% *w*/*v*) of sodium carbonate solution were added. The mixture was mixed well and left for 40 min at room temperature in the dark. Later, the reaction mixtures absorbance was measured at 725 nm using a spectrophotometer (Ultrospec 3100 pro, Amersham Biosciences, OR, USA) against a reagent blank. The phenolic content of the ginger was expressed as mg GAE (gallic acid equivalents) per gram dry extract.

#### 2.3.3. Determination of Total Flavonoid Content (TFC)

The TFC of ginger extracts were measured using the aluminum chloride calorimetric method with slight modification [25]. First, distilled water was added to the extract solution (100 µL) to make a volume of 2 mL. Then, the sodium nitrite solution (0.15 mL, 5% *w*/*v*) was added to the mixture. After 6 min incubation, 0.15 mL of 10% *w*/*v* aluminum chloride was added to the mixture and allowed to stand for 6 min at room temperature. Next, 2 mL of 4% *w*/*v* sodium hydroxide was added along with distilled water to make a volume of 5 mL. Finally, the reaction mixture was allowed to stand in the dark for 15 min and measured at 510 nm using a spectrophotometer. Rutin was used as the standard for the calibration curve, and the flavonoid contents were expressed as mg RE (rutin equivalents) per gram dry extract.

#### 2.3.4. Determination of AA Content 

The AA content was determined following the method of Klein and Perry [26]. First, the dried ginger extract (150 mg) was re-extracted with 10 mL (1%, *w*/*v*) metaphosphoric acid for 45 min at room temperature and filtered using Whatman filter paper (No.4). Then, the filtrate (1 mL) was mixed with 9 mL (0.005%, *w*/*v*) of 2,6-dichlorophenolindophenol (DCPIP), and the absorbance was measured immediately at 515 nm against a blank. Finally, the AA content was calculated based on the authentic L-AA calibration curve. The results were expressed in mg AA per g extract of ginger. 

#### 2.3.5. Ferric Reducing Antioxidant Power (FRAP) Assay

FRAP activity was measured according to the method of Pulido et al. [27]. Briefly, acetate buffer (300 mM, pH 3.6), TPTZ (2,4,6-tripyridyl-s-triazine) 20 mM in 40 mM HCl and FeCl_3_⋅6H_2_O (20 mM) were mixed at the ratio of 10:1:1 to obtain the working FRAP reagent. The extracts solution (30 µL) was then mixed with 900 µL of working FRAP reagent, and distilled water was added to the mixture to make a volume of 1 mL. The samples were incubated at 37 °C in a water bath for 30 min. Later, the resulting solution absorbance was measured at 593 nm against a reagent blank. The methanol solutions of FeSO_4_⋅7H_2_O ranging from 10 to 100 µM were prepared and used for the calibration curve. The equivalent parameter concentration is defined as the antioxidant concentration with a Ferric-TPTZ reducing ability equivalent to 1 mM FeSO_4_⋅7H_2_O and expressed as mmol Fe (II) equivalents per gram dry extract.

#### 2.3.6. Determination of Total Antioxidant Activity (TAA) by Phosphomolybdenum Assay

The TAA of ginger extracts was evaluated by the green phosphomolybdenum complex formation, according to the method of Prieto et al. [28]. An aliquot of 100 µL of extract solution was mixed with 3.6 mL of reagent solution (0.6 M sulfuric acid, 28 mM sodium phosphate, and 4 mM ammonium molybdate) in test tubes. Then, the tubes were capped and incubated in a water bath at 95 °C for 30 min. After the samples had cooled to room temperature, the mixture absorbance was measured at 695 nm against the reagent blank [28]. The standard calibration curve was plotted using different concentrations of AA solution (10–100 µg/mL), and the TAA were expressed as g AA (ascorbic acid equivalents) per gram dry extract. 

#### 2.3.7. Determination of 2,2′-Azinobis (3-Ethyl-benzothiozoline-6-sulfonic Acid) Radical Cation (ABTS^•+^) Decolorization

The free radical scavenging activity of ginger extract was determined using ABTS radical cation decolorization assay [29]. In this assay, disodium salt (ABTS^•+^) was dissolved in water to achieve a concentration of 7 mM. Then, the ABTS radical cation (ABTS^•+^) was produced by reacting the ABTS stock solution with 2.45 mM potassium persulfate (final concentration) and allowing the mixture to stand in the dark at room temperature for 12–16 h before use. Before the assay, the solution was diluted in ethanol and equilibrated at 30 °C to give an absorbance of 0.70 ± 0.02 at 734 nm. Next, 0.9 mL of the diluted ABTS^•+^ solution was added to 100 µL of extract solution and incubated at room temperature for 30 min in the dark. The Trolox standards (final concentration 0–15 µM) were used as an antioxidant standard, and the results were expressed as mmol TEAC (Trolox equivalent antioxidant capacity) per gram dry extract.

#### 2.3.8. Determination of 1,1-Diphenyl-2-picrylhydrazyl (DPPH) Radical Scavenging Activity

The capacity of extracts to scavenge 1,1-diphenyl-2-picrylhydrazyl (DPPH) radical was determined according to the method proposed by Sowndhararajan et al. [30]. Initially, ginger extracts at various concentrations were adjusted to 50 µL with methanol. Next, 950 µL of 0.1 mM DPPH methanolic solution was added and shaken vigorously. Then, the mixtures were allowed to stand for 20 min at room temperature in the dark to obtain stable absorption. Next, the DPPH radical reduction was measured at 517 nm, and the DPPH^•^ scavenging activity was determined using the following Equation (2):(2)$$\%\;\mathrm{of}\;\mathrm{DPPH}•\;\mathrm{inhibition}=\frac{\mathrm{Control}\;\mathrm{absorbance}-\mathrm{Sample}\;\mathrm{absorbance}\;}{\mathrm{Control}\;\mathrm{absorbance}}\times 100\%$$

The extract concentration that inhibited 50% of free radicals (IC_50_) was determined by plotting a graph of the inhibition percentage against the concentration using the linear regression analysis. Antioxidants that possess higher scavenging abilities have lower IC_50_ values.

#### 2.3.9. Statistical Analysis

The data obtained were analyzed using analysis of variance (ANOVA), and the Tukey test was used to evaluate the significant difference between mean values at the confidence level of 95% (*p* < 0.05). Then, Pearson’s correlation was carried out to identify the correlation between antioxidant activities and phenolic content. Each analysis was conducted in triplicates, and the data were analyzed in Minitab 16 software (Minitab Inc., State College, PA, USA) and represented as the mean of three independent experiments.

## 3. Results and Discussion 

The antioxidant properties of dried *Bentong* ginger were investigated based on several parameters, including TPC, TFC, AA, FRAP, TAA, ABTS^•+^ scavenging activity and DPPH^•^ radical inhibition percentage. More than one antioxidant assay was conducted in the current study because each antioxidant assay has its strengths and limitation. Thus, the data from the four assays increase the confidence of the observations, especially for the antioxidant activities. Fresh ginger samples were used as control and labeled as F. The four drying methods are labeled as S, O, V, Z to denote sun-, oven-, vacuum oven-, and freeze-drying, respectively. Meanwhile, the selected extraction solvents were hot water, aqueous ethanol (80%), and ethanol (100%) labelled as H, A, and E. The labeling of F-H refers to fresh ginger samples extracted with hot water, and Z-E refers to freeze-dried ginger samples extracted with ethanol.

### 3.1. The Yield of Different Extraction Methods

Figure 1 shows that the extraction yield of *Bentong* ginger was significantly (*p* < 0.05) affected by different drying methods. The highest yield was obtained from sun-drying, extracted with aqueous ethanol. In aqueous extracts, sun-drying increased by 6.20-fold more than fresh ginger, compared to freeze-drying (5.73-fold), vacuum-drying (5.34-fold) and oven-drying (2.85-fold). For ethanol extracts, the extraction yield had a similar trend with the aqueous extract, where the sun-drying method showed the highest yield (3.04-fold), followed by freeze-drying (2.70-fold), vacuum-drying (2.52-fold) and oven-drying (1.46-fold). When using hot water extract, oven-drying remained the method with the lowest yield at 2.76-fold. Vacuum-drying gave the highest extraction yield at 4.41-fold, followed by sun-drying (3.69-fold) and freeze-drying (3.68-fold). Overall, the sun-drying method and aqueous solvent offer the best extraction yield. 

The higher extraction yield of dried ginger is linked to the increased sample wall permeability because of the drying treatment. Drying causes the tissue to become brittle, thus resulting in the breakdown of cell structure during milling and subsequently the liberation of intracellular compounds into the solvents [31]. Furthermore, the breakdown of cellular constituent release bound cellular compounds, thus increasing the yield. Moreover, drying treatments increase sample porosity, increasing the solute and solvent diffusion rate and higher extract yield [32]. Intracellular spaces (pores) previously occupied by water are replaced by air or compressed due to shrinkage during the drying process [33].

Sun-dried (moderate temperature: 36 °C) samples recorded the highest extraction yield in this study. This finding is in agreement with a review by Rahman [34], which reported that samples dried at a low temperature (20 °C) experienced higher shrinkage, whereas, at high temperatures (50–80 °C) the samples exhibited limited shrinkage [34]. Similarly, Wang and Brennan [35] reported that potato tissue shrinkage was significantly higher at low temperature (40 °C) than at high temperature (70 °C). According to Aprajeeta et al. [33], shrinkage and porosity during drying change with simultaneous heat and mass transfer. Therefore, when a sample is exposed to an extreme condition, such as high temperature, it causes a rapid decrease in surface moisture and crust formation, creating an impermeable layer that may lead to lower yield during extraction [36].

In terms of extraction solvents, aqueous ethanol (80%, *v*/*v*) was the most efficient for extraction, followed by hot water and ethanol. Likewise, Yeh et al. [37] reported that aqueous extracts of Taiwan ginger had a higher extraction yield than ethanol extracts. In addition, Anwar et al. [38] found that 80% aqueous organic solvent was the best solvent for the highest extraction yield of *Brassica oleracea,* compared to ethanol and methanol. The outcomes may result from the higher solubility of other compounds, such as carbohydrate, protein, and terpene, leading to the superior performance of aqueous extraction compared to hot water and ethanol [39]. Nevertheless, Anwar et al. [38] and Do et al. [39] mentioned that the recovery of extraction yield might not translate to higher antioxidant activity. The aqueous solvents may solubilize a more extensive range of compounds from the plant, containing little or no antioxidant compounds. 

### 3.2. Comparison of TPC, TFC and AA Content between Samples

Table 1 demonstrates that dried ginger contains more TPC and TFC than fresh ginger. The drying process helps break down the cell wall in the food matrix, thus releasing more phenolic compounds. Alternatively, active enzymes in the fresh sample degraded the antioxidant compounds, resulting in a low concentration of total phenolic and flavonoids in fresh ginger [40]. The destructive enzymes could have been inactivated due to the low water activity in dried samples, retaining the antioxidants in dried extracts [31,40]. Tomaino et al. [41] postulated the formation of new compounds with potential antioxidant capacity from the drying process, where an increase in antioxidant compounds was observed. 

The highest TPC was obtained from Z-E (20.91 mg GAE/g extract) with an increment of 2.60-fold from the fresh ginger. Similarly, Gumusay et al. [14] stated that the freeze-drying method resulted in the highest TPC levels than other methods. The highest level of phenolic content in freeze-dried extracts might be due to the ice crystals formed within the plant matrix during freezing. This process may have amplified the cell wall structure disruption, thus allowing the release of phenolic compounds [42]. In addition, freeze-drying involves sample lyophilization, preventing the thermal degradation of compounds. Therefore, the low temperature (below 0 °C) might render the degradative enzymes inactive, increasing TPC [43].

Interestingly, S-E samples demonstrated a significantly high TPC value at 19.57 mg GAE/g extract, a 2.44-fold increase compared to the fresh sample. The slow moisture loss without pressure and temperature shock may have contributed to this finding. Furthermore, it may be a form of defense mechanism when plants are under stress, which is beneficial for the release of phenolic compounds [44]. Meanwhile, the TPC increment of ethanolic extracts from the vacuum- (2.04-fold) and oven-drying (1.95-fold) were not significantly different, which may be due to the decomposition of antioxidant components after exposure to thermal treatment at 60 °C. High temperatures create unfavorable conditions for phenol composition extraction, leading to compound destruction [44].

The extraction of TPC using aqueous ethanol gave a smaller increment that ranged from 1.06 to 1.66-fold for sun-, oven-, vacuum- and freeze-drying in ascending order. There were no significant differences between extracted samples subjected to sun-drying and aqueous ethanol from the control. Furthermore, hot water extraction was the least effective since there were no significant differences in TPC levels of all samples compared to the fresh. The rapid activation of polyphenol oxidase (PPO) activity in samples may likely have caused the decline in phenolic contents, where the enzyme was able to generate oxidation products and reduce the phenolic contents [6]. Moreover, the presence of water molecules in the extracts may have prevented the complete release of the essential volatile components of ginger [21].

Dried ginger extracted with ethanol showed the highest TFC level from the sun-drying (651.5 g RE/100 g extract, 12.25-fold), followed by freeze-drying (541.5 g RE/100 g extract, 10.18-fold), oven-drying (489.8 g RE/100 g extract, 9.21-fold) and vacuum-drying (429.8 g RE/100 g extract, 8.08-fold). The highest-level observations in sun-drying were probably caused by developmental changes and stress-induced responses to sunlight and heat. During sun-drying, metabolically active plants would be under stress due to moisture loss [15]. Generally, plants produce phenylpropanoid compounds, such as flavonoids, as a defense mechanism in response to biotic and abiotic stress, such as infection, water stress, high temperature, and disease attacks [40]. Consequently, TFC levels were significantly higher for all samples after sun-drying. Another explanation for the high flavonoid compounds in sun-drying is the ultraviolet-B (UV-B) radiation effect on plant secondary metabolite production. It has been reported that UV-B rays indirectly modulate the physiological aspects of the plant by inducing constitutive changes and chemical defenses. Thus, UV-B rays trigger plants to synthesize valuable secondary metabolites compounds [45]. 

For aqueous extracts, there were slight increases in TFC content for sun-drying (1.31-fold), vacuum-drying (1.80-fold), oven-drying (1.92-fold) and freeze-drying (3.71-fold). However, only the freeze-dried extract yield exhibited a significant increase (*p* < 0.05) compared to the fresh. The non-thermal drying operation may have been beneficial in flavonoid release since the compound is heat sensitive, and its retention is dependent on temperature and drying treatment. As such, sensitive antioxidants, such as flavonoids in the freeze-dried samples, were significantly preserved. Freeze-drying is considered one of the best processing methods in maintaining plant quality [42,46].

Extraction using hot water produced low and non-significant TFC increment in the samples, ranging from 1.40 to 2.00-fold for oven-, sun-, freeze- and vacuum-drying in ascending order. Antioxidant compounds in fresh and dried ginger samples were degraded due to high enzymatic activities in the presence of water [47]. 

Besides phenolic and flavonoid compounds, AA is also an essential nutrient in food. The drying methods in this study did not significantly affect vitamin C content, reflected by the inconsistent trend. An AA is naturally more labile than other antioxidants; thus, the lengthy drying process (approximately three days) might have lost unstable vitamin C [40]. In addition, fresh ginger contains low vitamin C levels that could have been entirely oxidized and lost during the drying process [14]. Therefore, a short drying time and lower temperature are recommended in maintaining the AA content in fresh produce [13]. Oboh et al. [11] found that ginger contains low vitamin C content, while Gumusay et al. [14] reported that the AA content could not be found in the ginger samples. 

In the present study, the phenolic and flavonoid contents of ginger extract aligned with those of the studies by Loganayaki et al. [24] and Zhishen et al. [25]. However, the current findings contradicted an earlier study on the phytochemical contents of Malaysian ginger by Ghasemzadeh et al. [48]. The discrepancies in study outcomes might be caused by the different maturity, processing, and agriculture practices of fresh ginger samples [49].

### 3.3. Comparison of FRAP Values between Samples

Figure 2 illustrates the FRAP of ginger dried using different methods and extracted with hot water, aqueous ethanol and ethanol. The values ranged from 273.57 to 4200.51 mmol Fe(II)/g extract for ethanolic extracts, 915.37 to 2983.36 mmol Fe(II)/g extract for aqueous extracts and 224.35 to 425.77 mmol Fe(II)/g extract for hot water extracts. 

For ethanolic extracts, FRAP values of the sun-dried sample were significantly higher (*p* < 0.05) with an increase of 15.35-fold, followed by oven-drying (13.44-fold), freeze-drying (13.09-fold) and vacuum-drying (9.55-fold). Likewise, Jayashree et al. [50] reported similar findings for sun-dried ginger samples, where the maximum volatile oil and gingerol contents were highly retained, compared to the cabinet tray drying method. Gingerol and shogaol are the major antioxidant compounds of ginger oleoresin, attributed to their potent reductive ability. Therefore, the higher reducing ability of sun-dried ginger may have been contributed by the higher liberation of antioxidant compounds.

For aqueous ethanol extracts, the FRAP values of sun-dried ginger demonstrated a slight increase (1.06-fold) compared to vacuum-drying (1.48-fold), oven-drying (1.98-fold) and freeze-drying (3.36-fold). However, the freeze-dried sample extracted using 80% ethanol was significantly higher (*p* < 0.05) than other drying methods. A similar trend was also observed for TPC levels of freeze-dried aqueous ethanol samples, higher than oven-, vacuum- and sun-drying methods. Thus, it was proven that freeze-dried ginger in aqueous ethanol solvent could maintain the antioxidant activity of dried samples. In a different study, Benjakul et al. [51] found that the antioxidant activities of freeze-dried seed extracts were higher than oven-dried water extracts. In contrast, the antioxidant activity of oven-dried pomelo peel extract gave higher FRAP values than freeze-drying in 80% aqueous ethanol solvent. 

Meanwhile, the FRAP values in Figure 2 from hot water extracts from all drying methods were only at 0.53–0.68-fold and were not significantly different from the control. Therefore, it is essential to note that sample antioxidant activity depends not only on drying methods, but also on the type of extraction solvent.

### 3.4. Comparison of TAA Values between Samples

Figure 3 shows the total antioxidant capacity of samples was highest in ethanol extracts, followed by aqueous ethanol and hot water.

For ethanol extracts, TAA increased significantly after being dried with sun-drying, showing the highest increment at 6.82-fold, followed by oven-drying (6.77-fold), freeze-drying (6.71-fold) and vacuum-drying (6.06-fold). As for aqueous ethanol extracts, TAA of the freeze-dried samples exhibited the highest increase at 2.97-fold, followed by ovendrying (1.74-fold), vacuum-drying (1.61-fold) and lastly, sun-drying (1.21-fold). In hot water extracts, the TAA of extracts post-drying increased slightly (approximately 1-fold) for all drying methods, except sun-drying (0.82-fold), suggesting a decrease in antioxidant activity. 

### 3.5. Comparison of ABTS^•+^ Scavenging Activity between Samples

The efficacy of the ABTS^•+^ scavenging activity of *Bentong* ginger treated with different drying methods and extraction solvent is shown in Figure 4. The assay measured the reduction in the radical cation as the percentage inhibition. Trolox, an analog of vitamin E, is a positive control inhibiting the radical cation formation in a dose-dependent manner.

Like other antioxidant activities, the ABTS^•+^ scavenging activity of ethanol extracts remained the highest, followed by aqueous ethanol and hot water. Figure 4 demonstrates that sun-dried extracts had the highest activity with a 3.51-fold increase, followed by oven-drying (3.1-fold), vacuum-drying (2.9-fold) and freeze-drying (2.7-fold). For aqueous ethanol extracts, the freeze-drying method showed the highest increase at 1.48-fold, followed by oven-drying (1.18-fold), sun-drying (0.93-fold) and vacuum-drying (0.93-fold). Meanwhile, the hot water extracts did not exhibit any antioxidant improvement but deteriorated with values less than 1-fold.

### 3.6. Comparison of DPPH^•^ Scavenging Activity between Samples

In the DPPH assay, the extracts act against scavenging activity to determine the IC_50_. Therefore, a low IC_50_ value indicates greater scavenging capacity and increased antioxidant activity. The current findings showed that the DPPH^•^ scavenging activity of ginger samples increased significantly after being dried, especially in ethanol and an aqueous solvent. The DPPH^•^ scavenging activity percentage of dried ginger compared to fresh ginger is illustrated in Figure 5.

All the dried samples showed excellent DPPH inhibition, except for hot water extracts. Ethanol extracts exerted the best free radical scavenging activity. Furthermore, sun-dried ginger samples revealed the highest radical inhibition (the lowest IC_50_ = 15.23 μg/mL), followed by oven-drying (IC_50_ = 22.10 μg/mL), freeze-drying (IC_50_ = 22.25 μg/mL), and vacuum-drying (IC_50_ = 24.89 μg/mL). Overall, there was a 93–95% increase in DPPH^•^ scavenging activity for dried ginger in ethanol solvent. These results are in line with those of Stoilova et al. [52], who studied the antioxidant activity of the alcohol extract of Vietnam ginger and found that the DPPH radical inhibition reached up to 90.1%. 

As for aqueous extracts, there was an increase in DPPH scavenging activity during the drying treatments. Freeze-dried samples showed significantly higher (*p* < 0.05) activity than fresh ginger. The increasing order of DPPH^•^ scavenging activity in aqueous ethanol extracts was as follows: freeze-dried (77%) > oven-dried (53%) > vacuum-dried (48%) > sun-dried (24%) > fresh (control). Meanwhile, the samples extracted with hot water exhibited the lowest DPPH scavenging activity and the highest IC_50_ values ranging from 509.14 to 255.06 μg/mL, indicating low antioxidant capacity. 

Thus, the present study demonstrated that the DPPH^•^ radical scavenging effect improved post-drying with the appropriate solvent. Similarly, the DPPH^•^ inhibition activity also depends on the extractable phenolic contents, such as reducing power activity (Table 1). 

### 3.7. Correlation between Different Antioxidant Properties

The antioxidant properties of phenolic and flavonoids are based on the structural relationship between different parts of their chemical structure. Natural polyphenols can remove free radicals, chelate metal prooxidants, reduce radicals, terminal oxidation chain reaction and inhibit oxidases [7,11]. The major secondary metabolites found in ginger are gingerols and shogaols that provide a ‘hot’ sensation in the mouth. These non-volatile compounds are largely responsible for the pharmacological activity of ginger powder, extracts and ginger oleoresins [1]. 

The significant increase in ginger reducing power due to drying aligned with an earlier report, where the drying process increased the antioxidant compounds in ginger [20]. The reducing power, as displayed by the ability of the ginger extracts to reduce Fe^3+^ to Fe^2+,^ is a potent antioxidation defense mechanism based on electron transfer. The reductive ability can be an antioxidation defense mechanism, possibly through the ability of the antioxidant compounds to reduce transition metals [27,40]. Therefore, the higher reducing ability of dried ginger extract led to an improved protective effect compared to fresh ginger, which may be related to the extractable antioxidant phytochemicals. The reducing power assay values were reflected in the corresponding concentration of electron-donating antioxidants [11]. 

In the phosphomolybdenum assay, the TAA of sun-dried in ethanol solvent agreed with the extractable phytochemicals in Table 1. The antioxidant compounds in plants act as a reductant that reduces molybdenum (VI) to molybdenum (V), and this redox-linked reaction was used to quantify the sample reductive ability. Like the ferric-reducing power, sun-dried ginger extracted with ethanol solvent displayed the highest reductive ability, corresponding to the higher retained antioxidant compounds that act as reducing agents and electron donors. 

Apart from reductive ability, ABTS^•+^ radical cation and DPPH^•^ free radicals are also used to evaluate the radical scavenging activity of plant extract. These assays are commonly used to screen the overall antioxidant capacity as hydrogen atom donors and quench the free radicals. The antioxidants of ginger donate hydrogen from the phenolic hydroxyl groups, thereby reducing the ABTS^•+^ and DPPH^•^ radicals from initiating or propagating further oxidation reactions [53]. Both assays are measured by reducing the radicals as the radical inhibition at 734 nm and 517 nm, respectively. 

The type of extraction solvent influences the extraction yield, extractable phytochemicals and antioxidant capacity of plant samples. In the present study, ethanol followed by 80% ethanol was found to be the most efficient solvents in extracting phenolic compounds of *Bentong* ginger. Variations in extractable phytochemicals among the solvents used may be attributed to the different polarities [39]. It has been observed that the extraction efficiency of the plant raw material increases with increasing solvent polarity. Ethanol is the most efficient and widely used solvent to extract antioxidants due to its polarity and reported in several studies [38,39,54]. In the case of a polar solvent, the dipole interaction forces facilitate hydrogen-bond formation between solute–solvent, which increases the solvation power, thereby enhancing the compound solubility [55]. 

Despite being a highly polar solvent, hot water was the least efficient solvent for ginger extraction. Since water is a polar solvent, the high surface tension may wash away the plant tissue hydrophilic compounds, such as hydroxyl (OH-) or carboxyl (COO-) groups [56]. Nevertheless, there was an improvement in antioxidant activity when ginger was extracted using water and ethanol (80% aqueous solvent). This outcome may be caused by the relative polarity of antioxidant compounds and the increase of dissolving power between water–ethanol solvent and solute molecules. Previously, it has been reported that water–ethanol mixtures are an excellent solvent combination to extract polar antioxidants when prepared in appropriate proportions [32].

Since ginger exhibited excellent antioxidant activity and is a source of natural antioxidants [5,11,20], the effect of drying methods on gingers’ antioxidant properties were studied. At present, drying treatment was observed to decrease the moisture content and enhance the antioxidant properties of the ginger. Furthermore, an increase in antioxidant capacity after drying has been attributed to the release of bound phenolic compounds brought about by the cell wall breakdown [31] and the formation of new compounds with potential antioxidant properties [41]. 

The present study revealed that sun-dried ginger showed the highest phytochemical contents recovery, reflected in the sample reducing power, TAA and radical inhibition of ABTS^•+^ radical cation and DPPH^•^ radical. Furthermore, the increased antioxidant contents of sun-dried ginger were possibly due to developmental changes and stress-induced responses of the plant due to sunlight exposure [57]. Moisture loss and UV-B exposure are stressors that might have contributed to the synthesis of antioxidant compounds to repair the damaged tissue and as a defense system from injury. 

For instance, the withering step of Chinese cabbage under sunlight (‘UV-B’ exposure) for two days demonstrated an increase in plant phenolic contents [45]. To explain the plant biochemical processes, environmental stressors, such as moisture loss and/or sunlight radiation, activated several enzymes in the biosynthetic pathway of polyphenols under sun-drying [45]. Moreover, it was observed that sun-drying has notable strengths over equipment-based methods [50,58,59,60]. For example, Orphanides et al. [58] recorded a higher phenolic content in sun-dried samples compared to microwave- and oven-drying. Similarly, Ti et al. [59] reported that sun-dried samples had higher reducing power and superoxide anion radical scavenging activity than oven-dried *Aronia melanocarpa* samples. In a Mustafa et al. [6] study, sun-dried ginger extracts also showed the lowest IC_50_ values in the production of nitric oxide (NO) in lipopolysaccharide (LPS)-stimulated Raw 264.7 cells compared to oven- and freeze-drying. This biological effect could confirm the beneficial health effects of sun-dried extracts by demonstrating the anti-inflammation action. 

In the present study, oven- and vacuum oven-dried samples performance was not significantly different. Nonetheless, oven-dried samples in both ethanol and aqueous solvent exhibited superior antioxidant activity in FRAP values, TAA and radical inhibition of ABTS^•+^ and DPPH^•^ compared to the vacuum-drying method. This finding may be due to the evaporation of volatile and semi-volatile phenolic compounds with water in the samples during vacuum-drying, leading to extensive antioxidants degradation [61]. Likewise, Akdas et al. [61] reported a greater loss of phenolic contents when the samples were vacuum-dried than oven-dried at the temperature of 55–75 °C [61]. The lower antioxidant activity in vacuum-dried samples might be caused by the long drying period. Antioxidant activity degradation increased with the drying period, concurring with Mbondo et al. [62] who also reported that vacuum-drying retained higher antioxidant capacity than oven-drying in a shorter period.

On the contrary, freeze-dried samples extracted using 80% aqueous solvent exhibited higher reductive ability and scavenging activity than ethanol samples, thus increasing extractable antioxidant phytochemicals during processing. The freeze-drying method boasted better final product quality, including sample stability, defined porous product structure, increased recovery of antioxidants, longer shelf life at room temperature, and reduced weight for storage [46]. Despite the advantages of freeze-drying and the existence of technology that could retain the nutritional composition of end products, the industrial acceptance of this processing method has been relatively slow. 

This study also revealed the antioxidant activity of fresh ginger than dried ginger. Fresh *Bentong* ginger possesses a high moisture content of 90.91% that prevents the complete release of the plant’s essential volatile components [21]. Apart from that, oxidative enzymes, such as polyphenol oxidase and peroxide in fresh ginger [13], may have degraded the antioxidant compounds. In contrast, oxidative enzymes in dried ginger samples were inactivated due to low water activity, thus retaining more antioxidant compounds in the extract [40]. 

The higher antioxidant retaining power of dried ginger than fresh ginger may potentially be due to the bioactive compounds of shogaols in the former that occur naturally during the drying process, inducing new compound formation with enhanced antioxidant properties [42]. The major constituents of fresh ginger, gingerols, decreased when subjected to drying, but there was an increase in the dehydrated form known as shogaols [63]. Guo et al. [64] studied the relationship of gingerol and shogaol contents as a potential antioxidant of fresh and dried ginger. It was found that shogaols exhibited superior antioxidant activities and are present in dried ginger, while they exist at low levels or are unidentified in fresh ginger.

In summary, drying minimizes the reduction of bioactive compounds by inhibiting the plant’s metabolic processes, preventing certain biochemical reactions that might alter the nutrition and composition of plant products, and inducing the formation of new antioxidant compounds with enhanced properties [40].

### 3.8. Correlation between Antioxidant Content and Antioxidant Activity

The Pearson’s correlation analysis was conducted to determine the relationship between the different antioxidant content and antioxidant activity of *Bentong* ginger (Table 2). 

A highly positive correlation was found between ferric reducing activity (FRAP) and antioxidant compounds of TPC and TFC with a corresponding correlation coefficient of r = 0.957 and r = 0.976, respectively. Furthermore, a highly positive correlation was observed between TAA and TPC (r = 0.964) and TFC (r = 0.966). Both correlations were statistically significant (*p* < 0.05) between antioxidant compounds and the reducing activity of the samples. The reductive ability of samples to reduce Fe^3+^/Fe^2+^ and phosphate-molybdenum (VI)/phosphate-molybdenum (V) were affected by the presence of TPC and TFC. The redox properties of these antioxidant compounds act as reducing agents via electron donation. According to Mojani et al. [65], gingerol and shogaol were the major phenolic constituents in ginger obtained during extraction. In addition, zingiberene and limonene distributed in variation among ginger varieties [50] could also reduce FRAP and TAA assay activity. 

Contrary to the FRAP and TAA assays results, a slightly weaker correlation between antioxidant compounds and free radical scavenging activity was observed. A Pearson correlation coefficient of r = 0.908 and r = 0.952 was observed between ABTS scavenging activity with TPC and TFC, respectively. This result indicated that the ABTS model was not strongly affected by phenolic contents. In addition, phenolic effectiveness depends on the molecular weight, the number of aromatic rings, and the nature of the hydroxyl group’s substitution rather than the specific functional group. Hagerman et al. [66] reported that high molecular weight phenolics, such as tannic acid, are better at quenching the ABTS^•+^ radical.

The IC_50_ DPPH activity model of the ginger extracts had the weakest correlation with TPC (r = −0.783) and TFC (r = −0.743). A negative correlation means the parameters were compared with IC_50_; the lowest IC_50_ gave a higher antioxidant activity. This correlation was statistically significant (*p* < 0.05) between antioxidant content and IC_50_ DPPH, implying that phenolic and flavonoid contents might not be the main bioactive compounds in ginger responsible for the DPPH free radical scavenging activity. The presence of other bioactive compounds in ginger, such as AA, glutathione, fiber and dietary fiber, could contribute to the antioxidant effects [11,14,37]. In addition, the differential response of the extracts in different antioxidant assays may be explained by the fact that the electrons/hydrogen transfer from antioxidants occurs at different redox potentials in different assay systems. The position and number of hydroxyl groups of phenolic compounds also influence antioxidant activities [67].

## 4. Conclusions

A suitable drying method and extraction solvent are essential to maximizing the phytochemical content and antioxidant activities of ginger extracts as a therapeutic agent and food ingredient. Results from the study showed that sun-drying using ethanol solvent produced dried ginger with high contents of flavonoids and exhibited the highest FRAP, total antioxidant activity and the strongest ABTS^•+^ radical cation and DPPH^•^ radical scavenging activity. Ethanol with high polarity produced the highest yield of phytochemical compounds and antioxidant activity amongst the three solvents used. Natural sunlight also preserved antioxidant properties with convenience at a low cost. Although aqueous ethanol has higher extraction yields, it gives lower levels of antioxidant activity than ethanol. In using aqueous ethanol extract, freeze-drying was the more suitable method. The use of sun-drying, which offers a green resource with much lower investment and maintenance cost compared to freeze-drying, will also encourage the use of solar thermal, photovoltaic panels and greenhouses technology in the agro-tech industry. 

## Figures and Tables

**Figure 1 foods-12-00178-f001:**
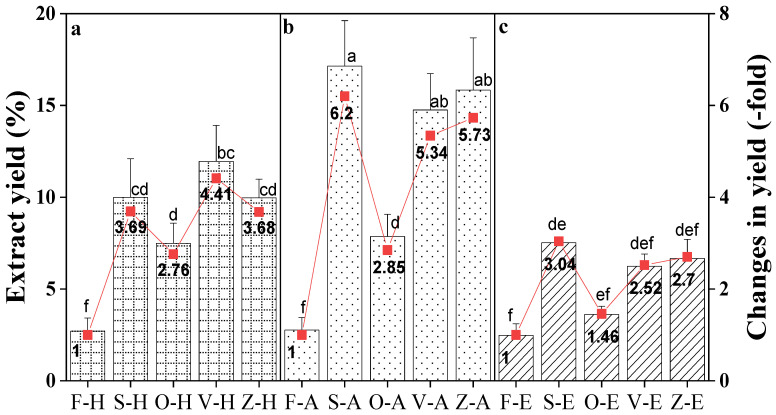
Extraction yield of *Bentong* ginger extracted using (**a**) hot water (H), (**b**) aqueous (A), and (**c**) ethanol (E) solvents via sun (S)-, oven (O)-, vacuum (V)- and freeze (Z)-drying methods. Fresh ginger (F) was used as the control. Changes were calculated based on the respective fresh ginger samples. Bars with different letters are significantly different. Sample F-H = Fresh ginger (Control) with hot water, and sample Z-E is freeze-drying with ethanol.

**Figure 2 foods-12-00178-f002:**
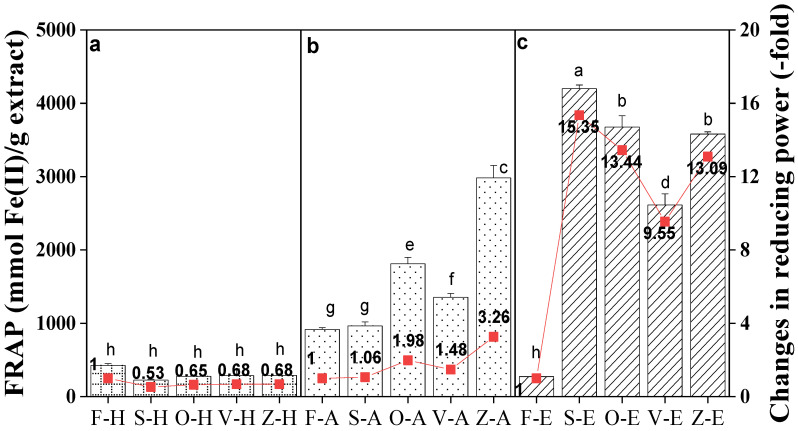
Reducing power of *Bentong* ginger extracted using (**a**) hot water (H), (**b**) aqueous (A), and (**c**) ethanol (E) solvents via sun (S)-, oven (O)-, vacuum (V)- and freeze (Z)-drying methods. Fresh ginger (F) was used as the control. Changes were calculated based on the respective fresh ginger samples. Bars with different letters are significantly different. Sample F-H = Fresh ginger (Control) with hot water, and sample Z-E is freeze-drying with ethanol.

**Figure 3 foods-12-00178-f003:**
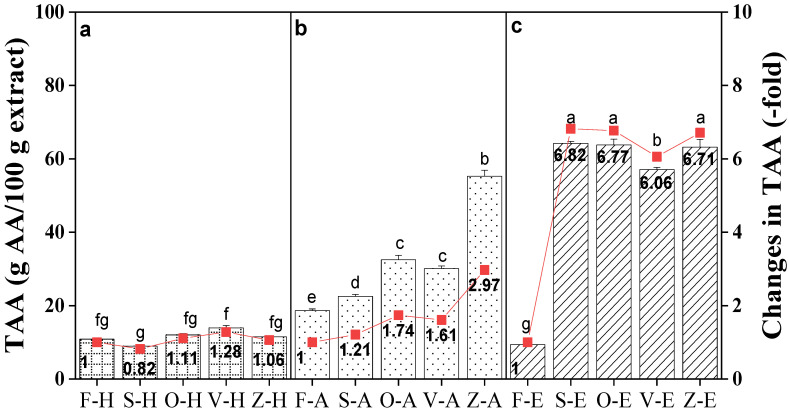
Total antioxidant activity (TAA) of *Bentong* ginger extracted using (**a**) hot water (H), (**b**) aqueous (A), and (**c**) ethanol (E) solvents via sun (S)-, oven (O)-, vacuum (V)- and freeze (Z)-drying methods. Fresh ginger (F) was used as the control. Changes were calculated based on the respective fresh ginger samples. Bars with different letters are significantly different. Sample F-H = Fresh ginger (Control) with hot water, and sample Z-E is freeze-drying with ethanol.

**Figure 4 foods-12-00178-f004:**
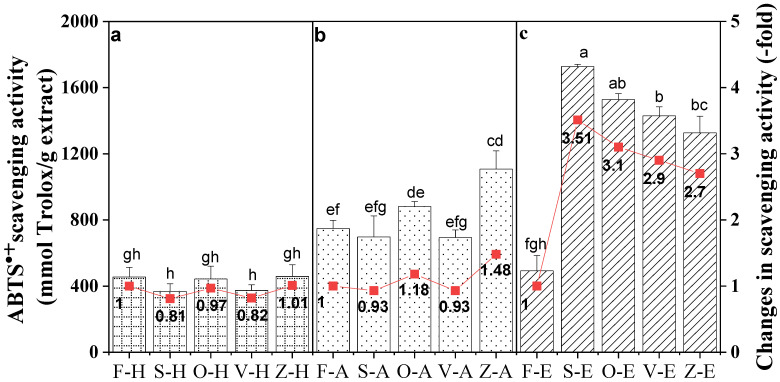
ABTS^•+^ scavenging activity of *Bentong* ginger extracted using (**a**) hot water (H), (**b**) aqueous (A) and (**c**) ethanol (E) solvents via sun (S)-, oven (O)-, vacuum (V)- and freeze (Z)-drying methods. Fresh ginger (F) was used as the control. Changes were calculated based on the respective fresh ginger samples. Bars with different letters are significantly different. Sample F-H = Fresh ginger (Control) with hot water, and sample Z-E is freeze-drying with ethanol.

**Figure 5 foods-12-00178-f005:**
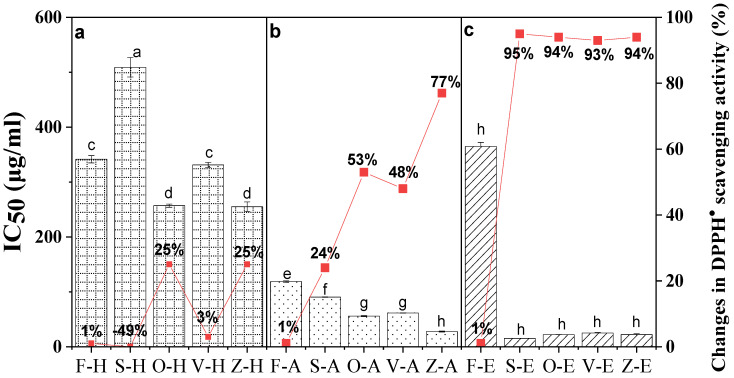
DPPH assay of *Bentong* ginger extracted using (**a**) hot water (H), (**b**) aqueous (A), and (**c**) ethanol (E) solvents via sun (S)-, oven (O)-, vacuum (V)- and freeze (Z)-drying methods. Fresh ginger (F) was used as the control. Changes were calculated based on the respective fresh ginger samples. Bars with different letters are significantly different. Sample F-H = Fresh ginger (Control) with hot water, and sample Z-E is freeze-drying with ethanol.

**Table 1 foods-12-00178-t001:** Total phenolic content (TPC) and total flavonoid content (TFC) of *Bentong* ginger extracted using hot water, aqueous ethanol, and ethanol solvents via sun-, oven-, vacuum- and freeze-drying methods. Changes were calculated based on the respective fresh ginger samples.

Sample	TPC (mg GAE/g Extract)	Changes in TPC (-Fold)	TFC (g RE/100 g Extract)	Changes in TFC (-Fold)
F-H	8.66 ± 0.18 ^g^	1.00 (Control)	41.50 ± 5.00 ^h^	1.00 (Control)
S-H	8.48 + 0.21 ^g^	−0.98	61.50 ± 13.23 ^gh^	+1.48
O-H	8.76 ± 0.44 ^g^	+1.01	58.20 ± 27.50 ^gh^	+1.40
V-H	9.07 ± 0.49 ^fg^	+1.05	83.20 ± 55.10 ^gh^	+2.00
Z-H	9.09 ± 0.09 ^fg^	+1.05	74.83 ± 2.89 ^gh^	+1.80
F-A	10.01 ± 0.21 ^ef^	1.00 (Control)	101.50 ± 5.00 ^fgh^	1.00 (Control)
S-A	10.63 ± 0.22 ^e^	+1.06	133.20 ± 41.90 ^efg^	+1.31
O-A	11.97 ± 0.46 ^d^	+1.20	194.80 ± 23.60 ^e^	+1.92
V-A	12.44 ± 0.20 ^d^	+1.24	183.17 ± 14.43 ^ef^	+1.80
Z-A	16.65 ± 0.14 ^c^	+1.66	376.50 ± 20.00 ^d^	+3.71
F-E	8.03 ± 0.06 ^g^	1.00 (Control)	53.17 ± 2.89 ^gh^	1.00 (Control)
S-E	19.57 ± 0.50 ^b^	+2.44	651.50 ± 40.00 ^a^	+12.25
O-E	15.63 ± 0.50 ^c^	+1.95	489.80 ± 43.10 ^bc^	+9.21
V-E	16.36 ± 0.47 ^c^	+2.04	429.80 ± 30.10 ^cd^	+8.08
Z-E	20.91 ± 0.61 ^a^	+2.60	541.50 ± 49.20 ^b^	+10.18

Mean values followed by different letters in a column are significantly different (*p* < 0.05). Sample F-H = Fresh ginger (Control) with hot water, and sample Z-E is freeze-drying with ethanol.

**Table 2 foods-12-00178-t002:** Pearson’s correlation between antioxidant content and antioxidant activity of *Bentong* ginger.

Antioxidant Activity	Antioxidant Content	
	TPC	TFC
Ferric reducing antioxidant power (FRAP) assay	r = 0.957	r = 0.976
Total antioxidant activity (TAA) assay	r = 0.964	r = 0.966
ABTS^•+^ radical cation scavenging assay	r = 0.908	r = 0.952
DPPH free radical scavenging assay	r = −0.783	r = −0.743

r = Pearson coefficient.

## Data Availability

The data presented in this study are available on request from the corresponding author.

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
