# Peer review of "Antioxidant Properties of Dried Ginger (Zingiber officinale Roscoe) var. Bentong"

_foods, 2023, doi:10.3390/foods12010178_

Round 1

Reviewer 1 Report

Presented for review manuscript with title: “Antioxidant properties of dried ginger (Zingiber officinale Roscoe) var. Bentong” is very interesting and well written.

 Presented manuscript is on high scientific level. The manuscript authors the introduction section presented the current state of knowledge on the experimental design. The topic is not a new but a very important, because show problems of quality of ginger fresh and dried by different technic methods.

The summary. Authors give a short presentation of manuscript with all aspects of content.

 The Introduction section includes all necessary information about examined objects and problems.

 Page 2, line 90: How many ginger rhizomes were use for experimental purposes? Please add this information to manuscript text.

 The aim of experiment formulated properly on the base of lack information in modern literature, as well filling in the gaps.

 Materials and methods 

All used analytical methods should have citation source. Please add it into manuscript text.

 Results

 All results present in a good way. Tables and Figures are clear and well visible. I don't have any comments to this section.

 The discussion section presents a good comparison of the obtained results with other results available in the data basis.

 Presented conclusions are corresponding with obtained results.

 General opinion:  After carefully manuscript reading, I think, that presented experiment is a valuable. In my opinion Manuscript should minor correction according to my points.

Author Response

Please see replies to comment as attached. Thank you.

Reviewer 2 Report

Authors did lots of basic works in the study and the research is meaningful for local food industry development. And the drying technology analysis also provided a guidance to the giger processing factory and ginger industry. Some critical issues should be clarified as follows:

1.       In the part of introduction, the review of drying technology occupied too many contents, I think it should be reduced.

2.       In the research method, I don’t think the total phenolic content is enough to explain the activities and obtain the object of this study, the detail information about the phenolic content should be detected using the method of such as UPLC-MS or other similar analysis methods.

3.       In the part of drying of ginger, “After the ginger slices were dried to 80 – 90% dry matter content”, here, did authors analyze the moisture content of samples using different drying technology? If the moisture is different in samples, the extracting yield might be inaccuracy.

4.       I don’t think authors give directly sufficient evidence for the antioxidant activities, the specific chemicals or polyphenol compounds mainly present in the raw material should be clarified.

Author Response

(The authors gave the same response as above.)

Round 2

Reviewer 2 Report

Most of issues has been clarified. It could be acceptable for publication.